# Can digital transformation promote green innovation in enterprises? the moderating effect of heterogeneous environmental regulations

Qiong Sun[1], Zhongsheng Wang[1], Yu Sun[1], Xiankai Huang[2]*

**1** Beijing Union University, Beijing, China, **2** Beijing Technology and Business University, Beijing, China

* huangxiankaibtbu@163.com

## Abstract

Currently, digital transformation is having various impacts on enterprises around the world, including the green innovation. However, the current literature on the relationship between digitalization and green innovation in enterprises is scarce. What is the relationship between them, and whether heterogeneous environmental regulation has mediating effects, are questions that are worth exploring. Using a sample of listed manufacturing enterprises in China, this paper empirically tests the impact of digital transformation on enterprise green innovation. The results show that: (1) Digital transformation has a significant positive impact on green innovation, including green innovation output and green innovation capability. (2) Diverse environmental regulation may have mediating effects of digital transformation's influence on green innovation. (3) After a number of robustness tests, the conclusions are still valid. This paper can provide a reference for developing green development strategies for manufacturing enterprises.

## 1 Introduction

Currently, digital transformation is having various impacts on enterprises around the world. For pollution-producing enterprises (especially manufacturing industries), digitalization of enterprises not only refers to the production efficiency of enterprises, but also refers to the greening of enterprises. The digitalization of enterprises will have an impact on technology development, and therefore on green technology. China is a typical example for such study.

In recent years, with the development of digital technologies such as big data, artificial intelligence, and cloud computing, digital transformation has played an increasingly important role in enhancing financial performance [1], achieving high-quality enterprise development [2], and alleviating environmental pressure [3,4]. With the nesting and application of digital technology, enterprises accomplish the transformation of production and operation mode and development mode, hence enhancing their total factor productivity [5] and output efficiency [6]. Some scholars proposed that the digital transformation can support enterprise innovation by influencing the organizational management efficiency, production and operation mode

**Funding:** This study was funded by the General Project of the Beijing Social Science Foundation (23GLB020). The funders had no role in study

design, data collection and analysis, decision to publish, or preparation of the manuscript.

**Competing interests:** The authors have declared that no competing interests exist.

transformation. As can be seen, the majority of current research on the digital transformation of enterprises focused on the economic and management implications. The effects of digital transformation on green innovation and environmental action are rarely discussed. Some scholars proposed that, digital technology can lower informational barriers, enhance the ability to acquire and improve resources, and foster enterprise green innovation [7]. Therefore, can digital transformation promote green innovation in manufacturing enterprises under environmental regulations? And what are the influence mechanisms? The answers can not only help to promote the economic development of manufacturing enterprises, but also push their green development.

In this study, we aim to establish a connection between digital transformation and green innovation. We employ an innovative and environmental regulatory perspective to develop a comprehensive theoretical model that examines the strategic implementation, ascending capabilities, and innovative outputs of green innovation in the context of digital transformation. This model also explores the intermediary role of green innovation capacity in regulating environmental regulations and heterogeneous environments. By unpacking the "black box" of digital technology's impact on green innovation processes, this research offers valuable insights for scholars interested in the intersection of digital transformation and environmental sustainability.

Our findings provide critical theoretical support for manufacturing enterprises seeking to achieve sustainable green development. They also offer important theoretical grounding and decision-making guidance for governments seeking to craft and implement green development policies that address environmental challenges posed by rapid industrial growth.

The organization of the paper is as follows: (1) In Section 2, a review of the relevant literature is presented; (2) Section 3 is dedicated to the presentation of the theory and hypothesis; (3) Section 4 details the study design, including the establishment of the model, the description of variables, and the description of data; (4) Section 5 contains the results of the empirical analysis; and (5) Section 6 provides the conclusion and recommendations.

## 2. Literature review

### 2.1 Digital Transformation and its effects on firm

Digital transformation can be interpreted as "digital technology empowers entity enterprise," which refers to the firm's use of a mix of information, computing, communication, and connection technologies, which help to enhance transaction costs, foster innovation potential, and increase commercial value [3]. Under the simultaneous influence of economic push and social push, enterprises are more likely to implement transformation strategies to achieve green and sustainable development Let's look back on the economic and social implications of the digital transformation in light of the two effects on boosting the green development of enterprises.

First, we review the economic effects of digital transformation. Digital transformation aids in increasing the effectiveness of resource allocation and integration, enhancing the mode of operation, lowering operating expenses, and ultimately raising the level of enterprise performance [8–10] Usage of digital technology by conventional sectors has the potential to spur innovation and alter the composition of factor inputs, resulting to scale and scope economies [11]. Digital transformation can significantly lower labor costs and management costs by enhancing business flexibility, process flexibility, and resource scheduling flexibility [3,12,13]. These changes are crucial in raising the asset utilization rate [14] and total factor productivity [15].

Second, we review the social effects of digital transformation. Current research on the social implications of digital transformation focuses mostly on the accountability behavior and carbon reduction behavior of businesses. Enterprise digitalization includes open, co-created, and

shared digital thinking, which has strong externality features and is significantly related to stakeholder relationship maintenance and the performance of corporate social responsibility [16]. Digital transformation can enhance the social responsibility performance of businesses by enhancing their capacity for real-time ecological environment monitoring and pollutant source identification and treatment [5]. Digital transformation can help businesses become more efficient and refined with their energy use, as well as controllable and predictable with their energy circulation [17], encourage enterprises to alter their primary mode of production that relies on fuel and oil consumption [18], and helping track their carbon emissions, hence reducing their carbon emission behavior [19].

## 2.2 Digital transformation and green innovation

Green innovation refers to the value creation activities that allow businesses to improve their environmental performance by enhancing or producing green technologies, processes, and products to lessen their negative impact on the environment [20,21]. As we can see from section 2.1, digital transformation leads to technological advances, drives down costs and economies of scale, all of which are theoretically conducive to green technological progress. But in reality, such relationship is more complex. As Cuerva et al. [22] pointed out, technological capabilities such as R&D just improve the conventional innovation but not necessarily push the green innovation, organizational and managerial factors play an important role in this. Clearly, digital transformation is something that can drive changes in technology, information, and the way companies work. That's why it's important for us to study digital transformation.

Social responsibility and environmental regulation are another vital term we should consider. Increase in environmental awareness can encourage enterprises to take the initiative to accept environmental responsibility and reduce environmental pollution through the use of green product innovation, green equipment innovation, and green process innovation [23]. Costs associated with environmental regulation can be mitigated by the innovation compensation effect in accordance with Porter's hypothesis, which would aid enterprises in implementing green innovation [24]. Some scholars found that there is a nonlinear relationship with temporal evolution between environmental regulation and green innovation rather than an absolute blockage or encouragement. Environmental protection investment of enterprises has a crowding out effect on green innovation investment in the short term, but the innovation compensation effect can compensate for this crowding out effect in the long term, thereby supporting green innovation of enterprises [25].

Overall, the impact of digital transformation on green innovation is likely to be multifaceted, which requires further assessment based on the previous literature.

## 2.3 Heterogeneous environmental regulations and its effects

As we review from the section 2.1 and section 2.2, digital transformation requires consideration not only of technical and economic factors, but also of social factors and intangible factors of the firms (e.g., organization, management, etc.). A very important instrumental variable for social factors here is environmental regulation. The current study concluded that, environmental regulation policy instruments can be classified as three categories of command control, market incentives, and public participation [26,27]. Among them: (1) The command control environmental regulation refers to the government's direct behavior in the pollution behavior of enterprises through law; (2) The market incentive environmental regulation refers to the enterprise pollution behavior through environmental tax, subsidies, economic stimulus, and other policies, etc.; (3) And the public participation environmental regulation refers to the public's participation in the oversight of environmental regulations. Some scholars had studied

the heterogeneous environmental regulation on green innovation. Jia et al. [28] found that the increased cost of enterprises resulting from heterogeneous environmental rules is offset by green innovation, hence encouraging enterprises to engage in green innovation. Therefore, environmental regulation and green innovation's influence mechanism should be considered in our analysis, which is ignored in the previous study.

## 3. Theory and hypothesis

### 3.1 Digital transformation and enterprise green innovation

**3.1.1 Digital transformation and enterprise green innovation output.** The adoption of digital transformation is a key factor in the high-quality development of businesses and also serves as a powerful engine for their adoption of green innovation. Some scholars had provided reference about how digital transformation drives enterprise green innovation: Pan and Faye [29] proposed that, digital transformation forced enterprises to focus on opportunities for green market development and cutting-edge technology, leading to the accumulation of knowledge resources and data capital for green innovation. Liu and Lu [30] argued that information system linked with digital technology enables enterprises to realize the process management of "green information acquisition-green information processing-green strategic decision-making," which significantly increases the efficiency of creative decision-making. Zhang and Long [31] found that, the digital system can accurately monitor the loss and lack of resources in each link of green innovation, prompting managers to allocate resources in a timely manner, effectively reduce resource mismatch, and improve the allocation efficiency of green resources. The study of Zhao [5] showed that, the use of digital transformation widens the social network of enterprises, enables them to reduce the Metcalf effect, lowers the marginal cost of innovation, and increases the return on green innovation. As can be seen from the previous study, this paper puts forward the following hypothesis:

$H_1$: Digital transformation helps to promote green innovation output of enterprises.

**3.1.2 Digital transformation and enterprise green innovation output.** Green innovation capability refers to the process by which enterprises integrate and reconstruct their resources in order to adapt to changes in their internal and external environments and accomplish green development objectives [32]. The success of green innovation tests not only the firms' technological strength [19], but also their dynamic ability to respond to changes in the social environment and modify actively [33]. Enterprises can use cutting-edge technology to carry out green innovation in accordance with the society and market. This will improve the success rate of their green innovation efforts. Therefore, the green innovation capability should concentrate on technological strength and green dynamic capability. In general, enterprises' ability to be environmentally friendly and dynamic is enhanced through digital transformation. Based on this, this paper puts forward the following hypothesis:

$H_2$: Digital transformation can help improve the green innovation capability of enterprises.

### 3.2 The mediating role of heterogeneous environmental regulation

As be seen in the review section, diverse environmental regulation may have mediating effects of digital transformation's influence on green innovation. (1) First, the influence of executive orders can have an impact on the digitization of companies. According to Ulas [34], digital transformation process and SMEs can be negatively influenced government interventions. Because digitization of enterprises is an internal activity, it requires consideration of revenue

costs and is a long-term project that is difficult to accomplish in the short term. If there are government's tasks, enterprises are likely to pursue short-term goals. Therefore, command-control environmental policy instruments may have negative mediating effect. (2) Second, the market is a major force driving the digitalization of the enterprise [35]. When confronted with environmental regulation pressure, enterprises with stronger green innovation capabilities may rapidly acquire market information and promote green innovation. Therefore, market-incentive environmental policy instruments may have positive mediating effect. (3) Third, social responsibility is an important part of modern business operations. Public engagement has a direct impact on companies and is positively linked to both digitalization and green technology progress [16,26,27]. Therefore, this paper puts forward the following hypothesis:

$H_3$: Heterogeneous environmental regulations have mediating effects of digital transformation's influence on green innovation.

## 4. Study design

### 4.1 Sample selection and data source

The fast growth of the manufacturing industry has resulted in an array of environmental issues, including air pollution, water pollution, and solid pollution. Promoting the green development of manufacturing enterprises will aid in achieving national macro-level carbon emission reduction goals. This paper chooses the manufacturing listed companies in China from 2010 to 2021 as the research sample in accordance with the industry code of Guidelines for the classification of Listed Companies in China (for Trial Implementation amended by China Securities Regulatory Commission in 2012). After eliminating the ST stock and * ST stock companies with missing important data and anomalous financial status, the dataset consists of 1,212 listed companies. All continuous variables are decreased by 1% to eliminate the influence of extreme values. Data on digital transformation is based on the enterprise annual report from 2010 to 2021. Green innovation data are sourced from the Tai'an (CSMAR) database, environmental regulation data are sourced from the Global Law and Regulations Network database and the China Environmental Statistics Yearbook, and other financial and corporate governance data are sourced from the Guo'an (CSMAR) database.

### 4.2 Variable definition

1. Explained variable
   Green Innovation (GI). An important indicator of an enterprise's level of green innovation is the quantity of green patent applications, which may be used to measure that level of green innovation. This paper refers to the reference [36,37], and uses green patent applications and access to measure enterprise green innovation.

2. Explanatory variable
   Digital Transformation (DT). In order to create the enterprise digital transformation index, this study draws on the previous work [31,38,39]. This paper uses the frequency of keywords linked to digital transformation in annual reports from businesses. Meanwhile, by referring to policy documents such as *National Intelligent Manufacturing Standard System Construction Guide (2015 edition)*, we further expand the feature word base of digital transformation and form the feature words of digital transformation as shown in Table 1.

**Table 1. The feature words of digital transformation.**

| Class | Feature word |
|---|---|
| Artificial intelligence technology | Artificial Intelligence, Business Intelligence, Image Understanding, Investment Decision Assistance Systems, Intelligent Data Analysis, Intelligent Robots, Machine Learning, Deep Learning, Semantic Search, Biometrics, Face Recognition, Speech Recognition, Authentication, Autonomous Driving, Natural Language Processing, AI Technology |
| Big data technology | Big Data, Data Mining, Text Mining, Data Visualization, Heterogeneous Data, Credit Investigation, Augmented Reality, Mixed Reality, Virtual Reality, Data Management, Data Network, Data Platform, Data Focus, Data Science, Data Control |
| Cloud computing technology | Cloud computing, flow computing, Cloud IT, Cloud ecology, cloud services, cloud platform, graph computing, memory computing, multi-party secure computing, brain-like computing, green computing, cognitive computing, fusion architecture, 100 million level concurrency, EB level storage, Internet of Things, information physics system |
| Blockchain technology | Blockchain, digital currency, distributed computing, differential privacy technology, and smart financial contracts |
| Digital technology application | Mobile Internet, industrial Internet, mobile Internet, Internet medical, e-commerce, mobile payment, third-party payment, NFC payment, intelligent energy, B 2B, B 2C, C2B, O2O, snatched, intelligent wear, intelligent transportation, intelligent medical, intelligent customer service, smart home, intelligent investment, intelligent environmental protection, smart grid, intelligent marketing, digital marketing, unmanned retail, Internet finance, digital finance, financial technology, quantitative finance, Information sharing, information management, Information integration, Information software, information system, Information network, information terminal, Information center, Information technology, networking, industrial information, industrial communication, automatic detection, risk control, automatic monitoring, automatic control, intelligent fault diagnosis |

1. Mediating variables
   Green innovation capacity (GIC) refers to a company's capacity to make efficient use of limited resources and rely on technological superiority. Some scholars have developed a system for assessing an organization's capacity for green innovation, taking into account a variety of factors such as green products, technologies, and innovations, and evaluating that capacity through a questionnaire survey. The evaluation standards are inconsistent and the empirical findings are impacted since the content of the questionnaire varies among industries. According to some scholars, R & D investment, as an innovation resource, can improve companies' ability for innovation and encourage green innovation in businesses as it increases [40]. After possessing the innovation resources, an enterprise's capacity for green innovation can also be gauged by the number of technicians who can fully utilize those resources. Therefore, the primary indicators of an enterprise's capacity for green innovation in this study are the R&D investment and the number of technical staff.

2. Regulated variable
   The practice of Bao and Guo [41] is cited in this paper as a way to gauge the effectiveness of command-based environmental regulation by looking at the total number of environmental administrative penalties that have been issued in each region. To measure market-driven environmental regulation, the natural logarithm of the regional sewage fee is utilized. The total number of complaints and visits/the total number of regional residents to measure the environmental regulation of public participation.

3. Controlled variable
   This paper selects enterprise age, enterprise size, return on assets, asset-liability ratio, and

**Table 2. Variables definition.**

| Variable type | Variable name | Variable symbol | Measurement Methods |
|---|---|---|---|
| Explained variable | Green Innovation | GI | Number of green patents |
| Explanatory variable | Digital Transformation | DT | Using python tool, text mining of digital transformation, and logarithmic the measured word frequency after 1 |
| Mediating variables | Green Innovation Capability | GIC | The log value of the enterprise R & D investment + the log value of the number of technical personnel |
| Regulated variable | Command-controlled environmental regulation | ERA | Number of environmental administrative punishment cases in each regionTo measure the command-controlled environmental regulation |
| | Market-incentive-type environmental regulation | ERB | Market incentive environmental regulation is measured by the natural logarithm of regional pollutant discharge fees |
| | Public participation type of environmental regulation | ERC | Public participation environmental regulation is measured by the total number of petition completed / the total number of regional population |
| Controlled variable | enterprise age | Age | Year of that year-Year of establishment + 1 |
| | Size | Size | The log value of the total assets of the enterprise |
| | return on assets | ROA | Net profit / Total assets |
| | asset-liability ratio | Lev | Total liabilities / Total assets |
| | The nature of ownership | Nature | State-owned enterprises as 1, non-state-owned enterprises as 0 |

ownership type as control variables based on the relevant literature that has already been published. Table 2 displays the specific variables.

## 4.3 Model setting

To test the aforementioned hypotheses, benchmark regression models (1) and (2) were first constructed to verify $H_1$, $H_2$. Then, the intermediary pathway test model (3) is constructed to verify $H_3$. Among them, α, β and γ represent the regression coefficient of each variable, Control represents the control variable, and ε represents the random error term:

$$GI = \alpha_0 + \alpha_1 DT + \sum Controls + \varepsilon_1 \qquad (1)$$

Test the relationship between digital transformation and enterprise green innovation through the model (1), if the coefficient of the independent variable DT in this model is $\alpha_1$Significance, it indicates that the relationship between digital transformation and enterprise green innovation is significant, so as to test the hypothesis that $H_1$.

$$GIC = \beta_0 + \beta_1 DT + \sum Controls + \varepsilon_2 \qquad (2)$$

Model (2) is used to test the relationship between digital transformation and green innovation capability. If the coefficient β1 of the independent variable DT is significant, it indicates that digital transformation has a significant impact on green innovation capability, so as to test hypothesis $H_2$.

$$GI = \gamma_0 + \gamma_1 DT + \gamma_2 GIC + \sum Controls + \varepsilon_3 \qquad (3)$$

Model (3) is used to test the mediating effect of the company's green innovation capability. Only if the coefficient γ2 of the mediating variable GIC is significant can the mediating effect be verified. At the same time, if the coefficient γ1 of the independent variable DT is not significant, the mediating effect is complete. If the coefficient γ1 of the independent variable DT is

also significant, the mediating effect is partially mediating, so as to test hypothesis $H_3$.

$$GIC = a_0 + a_1 DT + \alpha_2 ER + \alpha_3 DT \times ER + \sum Controls + \varepsilon \qquad (4)$$

$$GIC = c_0 + c_1 DT + c_2 ER + c_3 DT \times ER + b_1 GIC + b_2 GIC + \sum Controls + \varepsilon \qquad (5)$$

Among them, ER represents the moderating variable, which substitutes command-and-control environmental regulation (ERA), market incentive environmental regulation (ERB) and public participation environmental regulation (ERC) in regression. According to the moderated mediation effect test method of Wen and Ye [41], model (4) is firstly tested, focusing on the significance of the coefficient $a_1$ of digital transformation (DT) and the coefficient $a_3$ of the intersection term. Test model (5) again and observe the significance of $b_1$ and $b_2$. If $a_1$ and $a_3$ are both significant, then heterogeneous environmental regulation regulates the relationship between green innovation ability and green innovation, so as to verify $H_4$.

## 5. Empirical analysis

### 5.1 Descriptive statistical analysis

Table 3 shows the mean, standard deviation, minimum, and maximum values of the major variable, where:

1. The average value of GI is 35.58, the standard deviation is 85.18, the minimum value is 0 and the maximum value is 1079, suggesting that the level of green innovation across the sample enterprises varies considerably. From the mean bias to the minimum value, it can be determined that the green innovation level of the majority of sample enterprises is at a low level, but only a few sample enterprises are at a high level. This indicates that the distribution of green knowledge capital and green technology resources is unbalanced among manufacturing enterprises, and that there is a "gap between wealthy and poor" phenomena.

2. DT has a mean of 65.52, standard deviation of 142.52, minimum of 0 and maximum of 1546. Similar to the statistical results of the level of green innovation, the degree of digital transformation of the sample enterprises varies significantly, which may be related to the digital policies of regional governments and the agglomeration effect of associated industries. Regional variations may contribute to the high degree of DT observed values' dispersion. All other variables have standard deviations between 0 and 20, and, relative to the first two variables, their observed value distributions are relatively balanced.

**Table 3. Descriptive statistical results.**

| Variable | Sample size | Mean | Standard deviation | Minimum value | Maximum value |
|---|---|---|---|---|---|
| GI | 820 | 17.18 | 46.08 | 0 | 782 |
| GIC | 820 | 23.81 | 2.447 | 15 | 33 |
| DT | 820 | 41.97 | 61.08 | 0 | 560 |
| ERA | 820 | 8.76 | 1.189 | 1 | 10 |
| ERB | 820 | 12.69 | 1.346 | 7 | 14 |
| ERC | 820 | 8.87 | 0.79 | 7 | 10 |
| Age | 820 | 22.04 | 4.929 | 10 | 45 |
| Size | 820 | 22.28 | 1.221 | 19 | 27 |
| ROA | 820 | 3.80 | 7.90 | -57.43 | 66.40 |
| Lev | 820 | 41.90 | 18.41 | 1.50 | 97.19 |

**Table 4. The correlation coefficients of the main variables.**

|  | GI | DT | GIC | ERA | ERB | ERC |
|---|---|---|---|---|---|---|
| GI | 1 |  |  |  |  |  |
| DT | 0.295** | 1 |  |  |  |  |
| GIC | 0.401** | 0.190** | 1 |  |  |  |
| ERA | 0.042 | 0.094** | 0.064 | 1 |  |  |
| ERB | -.0081* | 0.015 | 0.067 | 0.507** | 1 |  |
| ERC | -0.043 | 0.025 | 0.064 | 0.397** | 0.718* | 1 |

Note: The lower left corner is the Pearson coefficient; * indicates p <0.1

* * indicates p <0.05, and

* * * indicates p <0.01.

## 5.2 Correlation analysis

The correlation coefficients for the key variables in this study are displayed in Table 4. The digital transformation and enterprise green innovation Pearson correlation coefficient is 0.094, which passed the 1% significance level test, indicating a positive association between the two. The Pearson correlation coefficient between digital transformation and the ability for green innovation, as well as between the ability for green innovation and enterprise green innovation, was 0.126 and 0.411, respectively. These values passed the 1% significance level test, indicating a positive correlation between digital transformation and the ability for green innovation as well as a positive correlation between the ability for green innovation and enterprise green innovation. This provides first support for the hypothesis provided in this work. Furthermore, the variables' correlation coefficient is essentially smaller than 0.4. The variance expansion factor measures all the explanatory variables, and the results reveal a mean VIF of 1.01 and a maximum value of 1.016, both significantly below the threshold of 10, indicating that the explanatory variables in this study do not have a significant multicollinearity issue.

## 5.3 Analysis of the regression results

In the field of digital transformation, enterprises are presented with new opportunities to enhance their comprehension of green innovation and seek external resources. This, in turn, propels their advancement in green innovation. Furthermore, from an economic perspective, a one standard deviation increase in the proportion of digital transformation leads to a 31.24% growth in the green innovation index of enterprises, while controlling for other variables. It is evident that digital transformation has a positive and significant impact on enterprise green innovation. This supports Hypothesis H1.

The regression analysis of Model (2) reveals that the regression coefficient of digital transformation on green innovation capability is 0.086, significant at the 1% level. This suggests that digital transformation not only enhances the technical strength and dynamic capabilities of enterprises but also contributes to the overall enhancement of their green innovation capabilities. This supports Hypothesis H2.

The results from Model 3 regression indicate that the coefficient of green innovation capability, the intermediary variable, is 0.284, significant at the 1% level. This suggests that the capacity for sustainable innovation has a significant positive impact on enterprise-level innovation. The stronger the enterprise's green innovation capability, the more effective it is in boosting their green innovation efforts. Additionally, the independent variable digital transformation has a coefficient of 0.054, significant at the 5% level (Table 5). This indicates that the

**Table 5. Results of the regression of the main variables.**

| Variable | GI | | GIC |
|---|---|---|---|
| | Model (1) | Model (3) | Model (2) |
| DT | 0.198*** | 0.178*** | 0.005*** |
| | (8.302) | (7.450) | (4.952) |
| Age | 0.094 | 0.209 | -0.026** |
| | (0.305) | (0.685) | (-2.171) |
| Size | 11.529*** | 5.465** | 1.385*** |
| | (8.595) | (3.033) | (26.389) |
| ROA | -4.864 | -18.962* | 3.220*** |
| | (-0.250) | (-0.980) | (4.237) |
| Lev | 16.441* | 10.558** | 1.344* |
| | (1.870) | (1.207) | (3.906) |
| Nature | 4.628 | 5.119 | -0.112 |
| | (1.435) | (1.610) | (-0.890) |
| GIC | | 4.378*** | |
| | | (4.955) | |
| Constant | -258.434*** | -226.413*** | -7.314*** |
| | (-9.207) | (-7.968) | (-6.659) |
| N | 820 | 820 | 820 |
| R$^2$ | 0.209 | 0.232 | 0.571 |

Note: * indicates p <0.1

** indicates p <0.05

*** indicates p <0.01, and the t-statistic is given in parentheses.

intermediary effect exists and that the improvement of firms' green innovation level is influenced not only by digital transformation but also by the intermediary function of green innovation capability. The mediating effect test does not contradict Hypothesis H1, and Hypothesis H3 is supported.

The regression analysis of model (4) reveals that the coefficients of DT are 0.347, 0.343, and 0.588, respectively. These values are significant at the levels of 5%, 10%, and 5%, respectively, indicating that digital transformation has a significant positive impact on the green innovation capacity of enterprises under various environmental regulations.

The intersection terms GIC×ERA and GIC×ERC have coefficients of 0.362 and -0.618, respectively, according to the regression results of model (5). However, these results are not significant, indicating that environmental regulations based on public participation and command-and-control do not significantly modify the relationship between enterprise green innovation and sustainable innovation capacity (Table 6).

One possible explanation for this observation is that command-and-control environmental regulations and public participation environmental regulations do not offer immediate benefits to businesses. Therefore, they may not cover the costs of innovation, limiting the potential for green innovation to have a significant impact on sustainable innovation capacity. This suggests that government control and public participation, while strong examples of environmental regulations, do not significantly enhance or hinder the impact of green innovation on sustainable innovation capacity.

The moderated mediation model tested in this study is supported by the significant coefficient of the intersection term GIC×ERB (-0.963 at the 5% level). This finding indicates that environmental regulation can moderate the mediating effect of green innovation capacity on digital transformation and enterprise green innovation. Specifically, at a high level of market-

**Table 6. Moderated mediation path test results.**

| Variable | GI | | | GIC | | |
|---|---|---|---|---|---|---|
| | Model (5) | | | Model (4) | | |
| DT | 0.173*** (6.861) | 0.215*** (9.626) | 0.209*** (9.149) | 0.004*** (4.315) | 0.005*** (4.937) | 0.005*** (4.875) |
| ERA | 1.117 (0.898) | | | 0.181*** (3.777) | | |
| ERB | | -3.867** (-3.899) | | | 0.137** (3.243) | |
| ERC | | | -2.797 (-1.602) | | | 0.250*** (3.453) |
| DT ×ERA | 0.008 (0.376) | | | 0.001 (0.366) | | |
| DT ×ERB | | -0.149*** (-8.160) | | | -0.001 (-0.417) | |
| DT ×ERC | | | -0.225*** (-7.326) | | | -0.001 (-0.106) |
| Constant | -237.310*** (-7.566) | -167.807*** (-5.661) | -199.314*** (-6.230) | -9.128*** (-7.658) | -9.148*** (-7.422) | -9.696*** (-7.507) |
| Controls | Yes | | | Yes | | |
| N | 820 | 820 | 820 | 820 | 820 | 820 |
| $R^2$ | 0.233 | 0.355 | 0.317 | 0.578 | 0.576 | 0.577 |

Note: * indicates $p < 0.1$

** indicates $p < 0.05$

*** indicates $p < 0.01$, and the t-statistic is given in parentheses.

oriented environmental regulation, the impact of green innovation capabilities on enterprise green innovation is magnified, providing partial support for H4.

## 5.4 Test of robustness

According to Wu et al. [36], the entire digital transformation index of the enterprise can be broken down into "underlying technology level" and "practical application level." To quantitatively evaluate digital transformation, we calculate the total word frequency of artificial intelligence technology, big data technology, cloud computing technology, and blockchain technology at the underlying technology level and perform logarithmic processing. At the practical application level, we employ the logarithm of the word frequency of specific digital keywords to assess the extent of digital transformation. The regression results continue to strongly validate the aforementioned conclusions, as presented in Table 7.

To reduce the sample size, we exclude enterprises with green innovation levels of 0 and digital transformation degrees and 0, respectively. We conduct regression analysis to determine whether enterprises that have not implemented green innovation or digital transformation will impact the empirical results. The regression results continue to strongly support the aforementioned conclusions, as shown in Table 8.

## 6. Conclusion and suggestion

### 6.1 Conclusion

In this study, we have selected Chinese manufacturing listed companies from 2010 to 2021 as our research samples. We have conducted comprehensive research on the mechanisms and impacts of digital transformation on the green innovation capabilities of manufacturing

**Table 7. Robust test: DT caliber decomposition.**

| | (1)<br>GI | (2)<br>GI |
|---|---|---|
| L. The underlying technology | 0.365***<br>(8.860) | |
| L. Digital technology application | | **0.180***<br>(4.781) |
| Constant | -266.848***<br>(-9.583) | -265.897***<br>(-9.219) |
| Controls | Yes | |
| N | 820 | 820 |
| R² | 0.218 | 0.166 |

Note: * indicates p <0.1

** indicates p <0.05

*** indicates p <0.01, and the t-statistic is given in parentheses.

enterprises. Furthermore, we have also delved into the mediating role played by this capability and the moderating effects exerted by diverse environmental regulations. Initially, we conducted theoretical analysis on the promoting effect of digital transformation on green innovation in enterprises, based on micro-level digital transformation indicators. Subsequently, we empirically tested the positive correlation between these two factors.

In order to assess green innovation capabilities, we employed indicators such as the number of R&D personnel and the proportion of R&D funds to main business revenue. The research findings indicate that digital transformation has the potential to enhance the green innovation capacity of enterprises and encourage further green innovation. Furthermore, the regulatory impact of market incentive environmental regulations on the "digital transformation green innovation capability green innovation" path has been established. These conclusions offer empirical evidence for understanding the high-quality development of enterprises and promoting the enhancement of green innovation capabilities and levels in manufacturing enterprises. It is imperative for the government to proactively strengthen market incentive environmental supervision policies, encourage manufacturing enterprises to pursue green innovation activities, and mitigate environmental issues caused by their rapid development.

**Table 8. Robust test with deleting some samples.**

| | (1)<br>GI | (2)<br>GI | (3)<br>GI |
|---|---|---|---|
| L. DT | 0.215***<br>(7.328) | 0.202***<br>(8.050) | 0.219***<br>(7.139) |
| | Excluding enterprises with a green innovation level of 0 | Excluding enterprises with zero digital transformation degree | At the same time, enterprises with green innovation level and digital transformation level are excluded |
| Constant | -300.102***<br>(-7.893) | -272.775***<br>(-9.078) | -314.897***<br>(-7.739) |
| Controls | Yes | | |
| N | 588 | 762 | 552 |
| R² | 0.207 | 0.211 | 0.208 |

Note: * indicates p <0.1

** indicates p <0.05

*** indicates p <0.01, and the t-statistic is given in parentheses.

## 6.2 Suggestion

According to the conclusion of this paper, the following enlightenment can be obtained:

1. Manufacturing enterprises must expeditiously pursue digital transformation in alignment with their unique resource endowments. This effort must aim to achieve breakthroughs in traditional production, operations, and management. By optimizing resource structures, reducing energy consumption, and curtailing cost waste, manufacturing enterprises can drive further enhancement of value creation and economic benefits.

2. A strong emphasis must be placed on cultivating new green innovation capabilities to foster green innovation within manufacturing enterprises. Research findings indicate that green innovation capability serves as a mediating factor between digital transformation and enterprise green innovation. Therefore, manufacturing enterprises must work to enhance their technological strength and dynamic capabilities to bolster green innovation capability. This approach will mitigate green innovation risks and improve the conversion rate of green innovation outcomes, ultimately enabling manufacturing enterprises to avoid long-term increases in innovation costs and achieve cost-effective innovation compensation effects.

3. The government must optimize market-based incentive policies aimed at promoting green innovation in manufacturing enterprises. It has been found that market-based incentive environmental regulation can effectively foster green innovation among enterprises with robust green innovation capabilities. To address environmental challenges like carbon pollution and water pollution, the government should seek to encourage green innovation among manufacturing enterprises through market incentives or other mechanisms. This approach serves as an effective way to drive green transformation and high-quality development within these enterprises.

## Author Contributions

**Data curation:** Qiong Sun.

**Funding acquisition:** Xiankai Huang.

**Resources:** Zhongsheng Wang.

**Writing – original draft:** Qiong Sun, Zhongsheng Wang.

**Writing – review & editing:** Yu Sun, Xiankai Huang.

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
