## [Decision Letter · Decision Letter 0]

2 Jan 2024

PONE-D-23-41651Can Digital Transformation Promote Green Innovation in Enterprises? The moderating Effect of Heterogeneous Environmental RegulationsPLOS ONE

Dear Dr. Huang,

Thank you for submitting your manuscript to PLOS ONE. After careful consideration, we feel that it has merit but does not fully meet PLOS ONE’s publication criteria as it currently stands. Therefore, we invite you to submit a revised version of the manuscript that addresses the points raised during the review process.

We look forward to receiving your revised manuscript.

Kind regards,

Han Lin

Academic Editor

PLOS ONE

Journal Requirements:

   "Supported by The Project of Cultivation for young top-motch Talents of Beijing Municipal Institutions (BPHR202203211), Innovation Centre for 

Digital Business and Capital Development of Beijing Technology and Business University (SZSK202237) and Academic Research Projects of Beijing Union University (SK10202201)"

   "Supported by The Project of Cultivation for young top-motch Talents of Beijing Municipal Institutions (BPHR202203211), Innovation Centre for Digital Business and Capital Development of Beijing Technology and Business University (SZSK202237) and Academic Research Projects of Beijing Union University (SK10202201)."

   "Supported by The Project of Cultivation for young top-motch Talents of Beijing Municipal Institutions (BPHR202203211), Innovation Centre for Digital Business and Capital Development of Beijing Technology and Business University (SZSK202237) and Academic Research Projects of Beijing Union University (SK10202201)."

6. In the online submission form, you indicated that "The datasets used and/or analysed during the current study available from the corresponding author on reasonable request." 

Additional Editor Comments:

The theoretical contributions could be better articulated, linking them back to the identified conversants and the research methods employed. This, in tandem with an explicit depiction of how the results impact our understanding of the research topic, would substantially elevate the study's contribution. It would be beneficial if the authors could identify the papers they aim to contribute to and delineate their novel addition to the existing discourse.

It might be helpful with new citations of up-to-date and high-quality papers on the subjects covered in manuscript and engaged the previous literature in this journal.

Also, the author(s) should carefully format the manuscript following the guidance of the journal, such as the headers, references, and so on. There are mistakes in the citation (reference), such as missing volume, issue and page.

The authors are recommended to examine and proof the language. In addition, there are some typos and grammatical errors that need further attention.

Reviewers' comments:

Reviewer's Responses to Questions

**Comments to the Author**

1. Is the manuscript technically sound, and do the data support the conclusions?

Reviewer #1: Yes

Reviewer #2: Yes

2. Has the statistical analysis been performed appropriately and rigorously? 

Reviewer #1: Yes

Reviewer #2: Yes

3. Have the authors made all data underlying the findings in their manuscript fully available?

Reviewer #1: Yes

Reviewer #2: Yes

4. Is the manuscript presented in an intelligible fashion and written in standard English?

Reviewer #1: Yes

Reviewer #2: Yes

5. Review Comments to the Author

Reviewer #1: This paper has a novel research theme and provides an important reference for the implementation of enterprise digital transformation strategy. In addition, this paper has made a solid review of the existing literature, and the theoretical analysis is clear and direct, but there are several contents that can be optimized as follows.

1. The content of the revelation part of the paper is repeated, so it is suggested to sort it out again.

2. At present, the marginal contribution of the article is too much, please briefly present the marginal contribution of the article.

Reviewer #2: Digital transformation is an important way for enterprises to achieve green innovation. This paper takes listed manufacturing companies in China as samples, empirically tests the impact of digital transformation on enterprise green innovation and its intermediate mechanism, and expands the relevant research on digital transformation. However, this article would be more complete if it could be improved in the following aspects.

1. Supplement the chapter arrangement of the paper at the end of the introduction.

2. Make clear the theoretical contribution of this paper compared with existing research.

3.The writing specification of the article should be further matched with the publication requirements，such as author format and references.

6. PLOS authors have the option to publish the peer review history of their article (what does this mean?). If published, this will include your full peer review and any attached files.

Reviewer #1: No

Reviewer #2: No

---

## [Author Response · Author response to Decision Letter 0]

15 Jan 2024

Journal Requirements

Response: Thanks for your comments. Modifications have been made in the article.

 "Supported by The Project of Cultivation for young top-motch Talents of Beijing Municipal Institutions (BPHR202203211), Innovation Centre for Digital Business and Capital Development of Beijing Technology and Business University (SZSK202237) and Academic Research Projects of Beijing Union University (SK10202201)."

Response: Thanks for your comments. This study was funded by the General Project of the Beijing Social Science Foundation (23GLB020).

The sponsor of the article is the corresponding author of the article

PLOS requires an ORCID iD for the corresponding author.

4. In the online submission form, you indicated that "The datasets used and/or analysed during the current study available from the corresponding author on reasonable request." 

We selected Chinese manufacturing listed companies from 2010 to 2021 as our research sample, which involves official data from many companies. We only conducted statistical analysis on the data, and the companies did not authorize the data, so it is inconvenient to provide it. Please understand!

Additional Editor Comments:

1. The theoretical contributions could be better articulated, linking them back to the identified conversants and the research methods employed. This, in tandem with an explicit depiction of how the results impact our understanding of the research topic, would substantially elevate the study's contribution. It would be beneficial if the authors could identify the papers they aim to contribute to and delineate their novel addition to the existing discourse.

Response: Thanks for your comments. Revised in the article, detailed in sections 5.1-5.4, conclusion and outlook.

2. It might be helpful with new citations of up-to-date and high-quality papers on the subjects covered in manuscript and engaged the previous literature in this journal.

Response: Thanks for your comments. Journal articles were cited in references 20, 24, and 29.

3. Also, the author(s) should carefully format the manuscript following the guidance of the journal, such as the headers, references, and so on. There are mistakes in the citation (reference), such as missing volume, issue and page.

Response: Thanks for your comments. The article has been extensively revised.

4. The authors are recommended to examine and proof the language. In addition, there are some typos and grammatical errors that need further attention.

Response: Thanks for your comments. The article has been extensively revised.

Reviewers' comments:

Comments to the Author

Reviewer #1: 

This paper has a novel research theme and provides an important reference for the implementation of enterprise digital transformation strategy. In addition, this paper has made a solid review of the existing literature, and the theoretical analysis is clear and direct, but there are several contents that can be optimized as follows.

1. The content of the revelation part of the paper is repeated, so it is suggested to sort it out again.

Response: Thanks for your comments. The conclusion and presentation have been rephrased, and duplicate content has been removed.

2. At present, the marginal contribution of the article is too much, please briefly present the marginal contribution of the article.

Response: Thanks for your comments. The marginal contribution is elaborated in the article, which is reflected in sections 5.2, 5.3, 5.4, and conclusion.

Reviewer #2: 

Digital transformation is an important way for enterprises to achieve green innovation. This paper takes listed manufacturing companies in China as samples, empirically tests the impact of digital transformation on enterprise green innovation and its intermediate mechanism, and expands the relevant research on digital transformation. However, this article would be more complete if it could be improved in the following aspects.

1. Supplement the chapter arrangement of the paper at the end of the introduction.

Response: Thanks for your comments. The concluding part of the introduction has been modified.

2. Make clear the theoretical contribution of this paper compared with existing research.

Response: Thanks for your comments. The conclusion and presentation have been rephrased and discussed in detail.

3.The writing specification of the article should be further matched with the publication requirements，such as author format and references.

Response: Thanks for your comments. Modifications have been made in the article.

---

## [Decision Letter · Decision Letter 1]

22 Jan 2024

Can Digital Transformation Promote Green Innovation in Enterprises? The moderating Effect of Heterogeneous Environmental Regulations

PONE-D-23-41651R1

Dear Dr. Huang,

We’re pleased to inform you that your manuscript has been judged scientifically suitable for publication and will be formally accepted for publication once it meets all outstanding technical requirements.

Kind regards,

Han Lin

Academic Editor

PLOS ONE

Additional Editor Comments (optional):

Reviewers' comments:

Reviewer's Responses to Questions

**Comments to the Author**

1. If the authors have adequately addressed your comments raised in a previous round of review and you feel that this manuscript is now acceptable for publication, you may indicate that here to bypass the “Comments to the Author” section, enter your conflict of interest statement in the “Confidential to Editor” section, and submit your "Accept" recommendation.

Reviewer #1: All comments have been addressed

Reviewer #2: All comments have been addressed

2. Is the manuscript technically sound, and do the data support the conclusions?

Reviewer #1: Yes

Reviewer #2: Yes

3. Has the statistical analysis been performed appropriately and rigorously? 

Reviewer #1: Yes

Reviewer #2: Yes

4. Have the authors made all data underlying the findings in their manuscript fully available?

Reviewer #1: Yes

Reviewer #2: Yes

5. Is the manuscript presented in an intelligible fashion and written in standard English?

Reviewer #1: Yes

Reviewer #2: Yes

6. Review Comments to the Author

Reviewer #1: (No Response)

Reviewer #2: (No Response)

7. PLOS authors have the option to publish the peer review history of their article (what does this mean?). If published, this will include your full peer review and any attached files.

Reviewer #1: No

Reviewer #2: No

---

## [Editor Report · Acceptance letter]

10 May 2024

PONE-D-23-41651R1 

PLOS ONE

Dear Dr. Huang, 

I'm pleased to inform you that your manuscript has been deemed suitable for publication in PLOS ONE. Congratulations! Your manuscript is now being handed over to our production team.

Kind regards, 

on behalf of

Dr. Han Lin 

Academic Editor

PLOS ONE